# Genetic Diversity and Geographical Distribution of the Red Tide Species *Coscinodiscus granii* Revealed Using a High-Resolution Molecular Marker

**DOI:** 10.3390/microorganisms10102028

**Published:** 2022-10-14

**Authors:** Hailong Huang, Qing Xu, Huiyin Song, Nansheng Chen

**Affiliations:** 1School of Marine Sciences, Ningbo University, Ningbo 315211, China; 2CAS Key Laboratory of Marine Ecology and Environmental Sciences, Institute of Oceanology, Chinese Academy of Sciences, Qingdao 266071, China; 3Laboratory of Marine Ecology and Environmental Science, Qingdao National Laboratory for Marine Science and Technology, Qingdao 266200, China; 4Center for Ocean Mega-Science, Chinese Academy of Sciences, Qingdao 266071, China; 5Department of Molecular Biology and Biochemistry, Simon Fraser University, 8888 University Drive, Burnaby, BC V5A 1S6, Canada

**Keywords:** red tide species, *Coscinodiscus granii*, mitochondrial genomes (mtDNAs), molecular marker, genetic diversity

## Abstract

Diatoms are responsible for approximately 40% of the global primary photosynthetic production and account for up to 20% of global carbon fixation. *Coscinodiscus granii* is a red tide forming species of the phylum Bacillariophyta that has been detected in a wide range of coastal regions, suggesting the possibility of the existence of high genetic diversity with differential adaptation. Common molecular markers including 18S rDNA, 16S rDNA, ITS, *cox1*, and *rbcL* do not provide sufficient resolution for distinguishing intra-species genetic diversity, hindering in-depth research on intra-species genetic diversity and their spatial and temporal dynamics. In this project, we aimed to develop molecular markers with high resolution and specificity for *C. granii*, attempting to identify different taxa of this species, which will set up a stage for subsequent functional assays. Comparative genomics analysis of the mtDNAs of *C. granii* strains identified a genomic region with high genomic variations, which was used to guide the development of a molecular marker with high resolution and high specificity. This new molecular marker, which was named *cgmt1* (*C. granii* mitochondrial 1), was 376 bp in size and differentiated *C. granii* samples collected in coastal regions of China into three different clades. Preliminary analysis of field samples collected in various coastal regions in China revealed that *C. granii* clades were almost exclusively found in the Bohai Sea and the north Yellow Sea. This newly developed molecular marker *cgmt1* could be used for tracking intra-species genetic diversity and biogeographic distribution of *C. granii* in different ecosystems.

## 1. Introduction

Diatoms (Bacillariophyta) are single-celled eukaryotic algae that serve as primary source of photosynthesis (and oxygen production) in the ocean. Diatoms are the most species-rich group of algae, with more than 200,000 estimated species, contributing up to 20% of global primary productivity [1,2,3]. *Coscinodiscus* (Coscinodiscophyceae, Bacillariophyta) is a species-rich genus with 174 taxonomically accepted species (Guiry and Guiry 2022), among which 49 species have been identified in various coastal regions in China [4]. Some *Coscinodiscus* species can form red tides that may cause serious damage to aquaculture through competitive utilization of nutrients or by causing extensive clogging of fishing nets and other equipment [5,6] and can cause hypoxia and bring substantial losses to fishery by generating a large amount of sediments [7]. As some *Coscinodiscus* species cannot tolerate cold temperature, their distribution indicates warming ocean temperatures and have been used to track global warming (https://sites.evergreen.edu/vms-spring/coscinodiscus/accessed on 1 June 2016).

*Coscinodiscus granii* L.F. Gough 1905 is a cosmopolitan *Coscinodiscus* species [8]. *Coscinodiscus granii* plays an important role in the marine ecosystem and has an important impact on the carbon pool because of its relatively large cell sizes and high carbon content [9]. *Coscinodiscus granii* has been found to form characteristic autumn red tides in the Baltic Sea and exist at high densities in spring and summer in the North European Seas [10,11]. In China, *C. granii* is commonly found in coastal waters and has been listed as the dominant phytoplankton species in the Bohai Sea [12,13].

Despite the wide existence of *C. granii*, few studies have been conducted on *C. granii* [14], little is known about the biogeographic distribution characteristics of *C. granii* and about its genetic diversity. Early research on diatoms including *C. granii* mainly focused on morphological characteristics observed using microscopy. The application of molecular biology methods and common molecular markers including 18S rDNA, 28S rDNA, ITS, *cox1*, and *rbcL* enabled quantitative analysis of phytoplankton [15,16,17]. While these molecular markers can be effectively used to distinguish species, they have limited power in resolving intra-species variations. In addition, accumulating evidence suggested the existence of high genetic diversity among red tide strains isolated from different geographical regions. For example, *Phaeocystis globosa* strains isolated from Hong Kong and Shantou, China were shown to have different toxicity during red tides, optimum growth temperature, and colony size [18]. As such, we expect that similar studies are needed to address *C. granii* intra-species diversity, which will be valuable for tracking ecosystemic functions and the formation red tides. We hypothesize that *C. granii* genetic diversity plays a role in its adaptation and red tide development. The goal of this study was to develop a molecular marker that has low intra-genome variation and high resolution for distinguishing different strains.

Mitochondrial genomes (mtDNAs), which carry many important genes related to cell respiration and metabolism, have been analyzed in diverse studies, including evolutionary studies, species identification, and speciation analysis [19,20]. Multiple mtDNA-based molecular markers including *cox1* and cytochrome b (*cob*), 16S rDNA have been commonly used for species identification. Molecular markers with higher resolution and specificity have been developed based on the comparative analysis of mtDNAs for many eukaryotic algae [18,21,22].

In this project, we aimed to develop a molecular marker with both high resolution and high specificity for distinguishing *C. granii* strains through comparative analysis of *C. granii* mtDNAs. Taking advantage of the recently constructed mtDNA [9] as the reference sequence, we identified intra-species genome variations (GVs) among six *C. granii* strains. We also tested the applicability of this new molecular marker in probing *C. granii* genetic diversity in the coastal waters of China.

## 2. Materials and Methods

### 2.1. Strain Isolation, Culturing, and Preservation

Six candidate *C. granii* strains were isolated in water samples collected during expeditions in coastal waters of Jiaozhou Bay, China (Figure 1 and Table 1). The expeditions, which were carried out during August and November in 2020 on research vehicle “Innovation”, were supported by the Jiaozhou Bay National Marine Ecosystem Research Station, Institute of Oceanology, the Chinese Academy of Sciences. These strains were first identified using light microscopy (Zeiss, Oberkochen, Germany). Morphological characteristics of these strains accorded with the description of *C. granii* [23]. Single cells were identified by morphology and picked up by pipetting individual cells using a micropipette under an inverted light microscope, followed by multiple washes with sterile culture medium before transferring each single cell to 24-well polystyrene cell culture plate. The strains were cultured in L1 medium (including 1‰ volume fraction Na_2_SiO_3_·9H_2_O and a mixture of 1% streptomycin liquid 100×, Solarbio, Beijing, China) with autoclaved and sterile filtered seawater. The cultures were maintained at 19–20 °C, with an irradiance of about 68 μmol photons m^−2^ s^−1^ in a photoperiod cycle of 12 h L/12 h D. Vegetative cells of six cultures were harvested for sequencing. Identification of these *C. granii* strains were based on both morphological features and molecular characterization using full-length 18S rDNA and ribulose-1, 5-bisphosphate carboxylase (*rbcL*) genes [21]. Phylogenetic tree was based on maximum likelihood (ML), using by MEGA7 [24].

### 2.2. Field Sampling and Initial Preparation

Field samples were collected from 66 sampling sites from coastal waters of China, including the Bohai Sea (October 2019), the North Yellow Sea (October 2019), the Changjiang Estuary (March 2019), the East China Sea (September 2019), the Pearl River Estuary (June 2019), the Beibu Gulf (January 2019), and the Jiaozhou Bay (March 2021). Details of the sampling sites were shown in Figure 2 and Appendix A. At each sampling site, 1000 mL surface water was filtered using mesh (Hebei Anping Wire Mesh Co., Ltd., Shijiazhuang, China) to remove large, suspended solids, zooplankton, and phytoplankton, followed by a second filtration through a 0.2 μm polycarbonate membranes (Millipore Corporation, Billerica, MA, USA) using a vacuum filtration pump. The filter membranes were transferred in tubes and were then snap-frozen in liquid nitrogen for brief storage for a few weeks.

### 2.3. DNA Preparation and Genome Sequencing

For cultures of *C. granii* strains, algal cells at exponential growth phase were transferred to a 50 mL centrifuge tube. Then, the algal cells were obtained by centrifugation (12,000× *g*, 6 min) and stored in a 1.5 mL EP tube for storage in liquid nitrogen for subsequent DNA extraction. Total genomic DNA from cultures and field samples were extracted with DNAsecure Plant Kit (Tiangen Biotech, Beijing, China) according to manufacturer’s instructions and quantified using a NanoDrop One spectrophotometer (Labtech International Ltd., Uckfield, UK). Genomic DNA samples of six *C. granii* strains were prepared for genome sequencing. The sequencing libraries were prepared by using the NEB Next^®^ Ultra™ DNA Library Prep Kit for Illumina (NEB, Ipswich, MA, USA). The PCR products were purified using AMPure XP system (Beckman Coulter, Beverly, MA, USA), and libraries were analyzed for size distribution by NGS3K/Caliper and quantified using real-time PCR (Qubit^®^3.0 Flurometer, Thermo Scientific, Waltham, MA, USA). After qualification, the libraries were sequenced using a 2 × 150 bp Illumina NovaSeq 6000 platform (Illumina, San Diego, CA, USA) at Novogene (Beijing, China), yielding about 5 Gb sequencing data of paired-end reads with 150 bp in length.

### 2.4. Filtering and Assembly of Sequencing Data

Raw data were filtered into clean data with Fastq following a series of quality control (QC) procedures as previously described [25]. Here, briefly, the raw data processing steps included: (1) removing reads with >10 nt aligned to the adapters; (2) removing reads with ≥10% unidentified nucleotides (N); (3) removing reads with >50% bases having Phred quality < 5; and (4) removing putative PCR duplicates generated by PCR amplification in the library construction process.

The filtered reads were assembled into scaffolds with SPAdes (v3.14) [26], Platanus-allee (2.0.2) [27], and ABySS (2.1.5) [28] with default parameters. With the mtDNA of *Thalassiosira pseudonana* (NC_007405) serving as references, scaffolds corresponding to mtDNAs of *C. granii* were identified using BLAST with default parameters. Analysis of the resulting scaffold using MEGA 7.0 (v7.0) and DOTTER (v4.44.1) to estimate whether sequences at the ends achieved overlap. Subsequently, the obtained draft mtDNA sequences were checked by aligning sequencing reads against the mtDNA using the MEM algorithm of BWA v0.7.17. The results of alignments were extracted with SAMtools (1.9) and visualized with IGV v2.8.12 [18]. According to alignments, assembly errors were corrected and N regions were replaced in the draft mtDNA. Molecular markers including full-length 18S rDNA, 16S rDNA, ITS, and *rbcL* were assembled with SPAdes. Finally, the assembly results of the mtDNA and molecular markers of the *C. granii* strain (CNS00554) were all validated with BWA and IGV. The circular mtDNA (CNS00554) of *C. granii* was 34,970 bp in size (GenBank accession number: MW435847). The annotation of protein coding genes (PCGs), transfer RNA (tRNA) genes, ribosomal RNA (rRNA) genes, and open reading frames (*orf*s) was conducted using Open Reading Frame Finder (ORF finder) with SmartBLAST and BLASTP, tRNAscan-SE, and MFannot [9].

### 2.5. Phylogenetic Analysis and Synteny Analysis

A total of the 27 shared protein-coding genes (PCGs), including *atp6*, *8*, *9*; *cob*; *cox1*, *2*, *3*; *nad1-7*, *4L*, *9*; *rpl2*, *13*, *14*, *19*; *rps3*, *4*, *8*; and *tatC*, from 34 publicly available diatom mtDNAs and two species of Ochrophyta, were first individually aligned using MAFFT [29] and then trimmed using trimAL [30] with default parameters: gt = 1, and all amino acid sequences were concatenated using Phyutility v1.2.2 [31]. Mitochondrial genes of two species *Sargassum fusiforme* (KJ946428) and *Sargassum muticum* (KJ938301) in Ochrophyta were selected as out-group taxa. The maximum likelihood (ML) phylogenetic tree was constructed using IQ-TREE (v1.6.12) [32]. Bootstrap analysis was performed using the ultrafast bootstrap approximation with 1000 replicates. In this study, *p* value was 0.01 after the ILD test, which indicated that sequence concatenation did not affect phylogenetic accuracy [19]. Comparison of mtDNAs of *C. granii* and *C. wailesii* was shown in the CIRCOS (v0.69) [21]. The order and content of genes in the mtDNAs of *C. granii* and *C. wailesii* were also displayed by Microsoft Excel 2019.

### 2.6. Developing Molecular Markers with High Resolution and High Specificity Based on C. granii mtDNAs

To search for GVs, we aligned Illumina reads of the six *C. granii* strains against the mtDNA of the reference strain CNS00554 using BWA (v0.7.17) with default parameters. Alignment results were screened using SAMtools with default parameters, and single-nucleotide variants (SNVs) with homozygous support >85% were called using VarScan (v2.4.4) [33]. The nucleotide diversity (Pi) values of *C. granii* mtDNAs were evaluated. The 400 bp (the length was appropriate for metabarcoding projects using Illumina DNA sequencing platform) sliding windows in the mtDNA of CNS00554 for SNVs were visualized using the R package circlize (v0.4.11) [34]. Primer Premier 5.0 was used to design the primers (*cgmt1*-F: TGGTGGGGAGGTTCTGTT; *cgmt1*-R: TTAAGCCTAAGTAGAGTTGA) of the novel high-resolution and high-specificity molecular markers *cgmt1* (*C. granii* mitochondrial 1). The primers were synthesized by Sangon Biotech Co., Ltd., Shanghai, China. In order to verify the specificity and resolution of the designed primers, PCR amplification experiments were carried out. The PCR mixture contained 2 μL of each primer *cgmt1*-F and *cgmt1*-R (final concentration 200 nM each), 25 μL 2 × Taq Mastermix (Tiangen, China), 50 ng of template DNA and added ddH_2_O water to 50 μL volume. The PCR protocol was 94 °C for 3 min, 32 cycles of 30 s at 94 °C, 30 s at 55 °C, and 50 s at 72 °C, and a final extension at 72 °C for 10 min.

### 2.7. Genetic Diversity and Biogeographic Distribution Analyses Based on the Molecular Marker

With the high-resolution and high-specificity molecular marker developed in this project, we explored *C. granii* genetic diversity by examining amplification results of the molecular marker in 66 environmental samples collected from expeditions to the Bohai Sea, the North Yellow Sea, the Jiaozhou Bay, the Changjiang Estuary, the East China Sea, the Pearl River Estuary, and the Beibu Gulf (Figure 2 and Appendix A). The environmental samples were PCR amplified and cloned before Sanger sequencing. The obtained forward and reversed fragments were assembled by ContigExpress (Vector NTI Suite 6.0, Invitrogen). Sanger sequencing results were aligned using MAFFT followed by editing using MEGA 7.0. The ML phylogenetic tree was constructed by the method mentioned above.

## 3. Results

### 3.1. Morphological and Molecular Identification of C. granii Strains

Six candidate *C. granii* strains (CNS00554, CNS00613, CNS00614, CNS00741, CNS00746, and CNS00749) were collected in the Jiaozhou Bay. The cells of *C. granii* were easily identified in girdle view due to its wedged shape. The cell lengths of *C. granii* were about 60–300 μm and height was about 30–180 μm. In addition to morphological identification, this study used two common molecular markers (full-length 18S rDNA and *rbcL*) for further identification. The six strains were clustered well with *C. granii* sequences reported previously in the maximum likelihood (ML) phylogenetic trees based on the marker full-length 18S and *rbcL* (Figure 3). The results of phylogenetic analysis indicated that these six strains were all *C. granii*.

### 3.2. Phylogenetic Analysis and Synteny Analysis of C. granii mtDNAs

The complete mtDNA (GenBank accession number: MW435847) of *C. granii* (strain CNS00554) has a circular genome of 34,970 bp in size, encoding 60 genes (Figure 4). The ML phylogenetic trees constructed using 27 common PCGs shared by mtDNAs of 34 Bacillariophyta and two Ochrophyta species as outgroup taxa indicated that 34 species in Bacillariophyta formed a single clade-the Bacillariophyceae. The Mediophyceae and Coscinodiscophyceae were both paraphyletic assemblages. The phylogenetic tree also showed that the *C. granii* (MW435847) clustered with *C. wailesii* (MW122841) with robust support (Figure 5). Synteny comparison of mtDNAs of *C. granii* and *C. wailesii* revealed the order of genes in the mtDNAs of the two species were generally similar, except for the rearrangements (translocation and inversion) of a few gene blocks, including *trnP(ugg)*-*trnY(gua)*-*rps11* and *atp8*-*trnA(ugc)* (Figure 6).

### 3.3. Defining a Molecular Marker with High Resolution and High Specificity for Distinguishing C. granii Strains

To develop molecular markers for tracking genetic diversity of *C. granii* using Illumina DNA sequencing, the length of the molecular markers should ideally be between 350 and 400 bp. Comparative analysis of the six *C. granii* mtDNAs identified a 400 bp-window with dense variations (Figure 7A). Furthermore, sliding window analysis of nucleotide variability (Pi) was used by complete mtDNAs of six *C. granii* strains (Figure 7B). In the end, we identified a genomic region containing 6 SNVs in mtDNA of *C. granii* strain CNS00554 (position: 5649–6048 bp). According to the optimization principle of primer software design, the amplification primers were designed in the region of 5749–6248 bp. Phylogenetic analysis using this small region suggested that it could be used to effectively distinguish these strains as molecular marker. Six strains of *C. granii* were divided into three clades based on this region (Figure 7D), which achieved similar resolution as the complete mtDNA (Figure 7C). The high resolution of molecular marker, which we named *cgmt1*, was 397 bp in size.

To test the specificity of newly developed molecular marker *cgmt1*, two analyses were carried out as the follows. First, we carried out BLAST searches of *cgmt1* in NCBI NT database, which showed low similarity (less than 29%) to sequences of other species, suggesting high specificity to *C. granii*. Second, we carried out PCR assays using primers designed against *cgmt1* and DNA preparations extracted from different 14 representative eukaryotic algae species, including *Amphidinium carterae*, *Alexandrium tamarense*, *Karenia mikimotoi*, and *Prorocentrum donghaiense* of the phylum Dinoflagellata, *Chattonella marina*, *Aureococcus anophagefferens*, and *Heterosigma akashiwo* of the phylum Ochrophyta, *Skeletonema costatum*, *Thalassiosira weissflogii*, *Chaetoceros curvisetus*, *C. wailesii*, and *Coscinodiscus* sp. of the phylum Bacillariophyta, *Phaeocystis globosa* and *Isochrysis galbana* of the phylum Haptophyta. We also included DNA preparations extracted from six *C. granii* strains as positive control. While most of these 14 representative eukaryotic algae species showed completely negative results, some showed some but greatly reduced signals. (Appendix A).

### 3.4. Probing C. granii Genetic Diversity and Geographical Distribution Using cgmt1

To evaluate genetic diversity and explore the biogeographical distribution of *C. granii* strains, we PCR amplified *cgmt1* from 66 environmental samples collected from various coastal areas of China. Among these 66 PCR amplicons, nine showed strong signals, which were further analyzed by cloning and sequencing. Sequencing results revealed that *C. granii* was mainly detected in the Bohai Sea and the North Yellow Sea (Figure 2). Phylogenetic analysis based on *cgmt1* sequence from 6 different strains of *C. granii* and 55 sequencing results from 9 positive environmental samples showed high genetic diversity (Figure 8). The phylogenetic analysis revealed at least three clades. Clade 1 included sequences of environmental samples from the Yellow Sea and the Bohai Sea, as well as sequences of strains from the Jiaozhou Bay. Clade 2 contained only sequences from the Jiaozhou Bay. Clade 3 contained sequences of environmental samples collected from the Bohai Sea and strains from the Jiaozhou Bay. The results indicated that the molecular marker *cgmt1* had high resolution, and that it could be used for tracking genetic diversity of *C. granii* in the field studies.

## 4. Discussion

*Coscinodiscus granii* has a worldwide distribution, suggesting adaptation to a wide range of temperatures [11] and salinities and possibly high genetic diversity. *Coscinodiscus granii* has been reported to form dense red tides, usually present great abundance in numerous ocean regions around the world, including the Bohai Sea and the Yellow Sea of China. However, current methods including morphology-based methods and common molecular marker-based methods have been inadequate for uncovering intra-species genetic differences. Common molecular markers including 18S rDNA, 28S rDNA, *rbcL*, and ITS have been demonstrated to be ineffective in resolving *Phaeocystis globosa* intra-species genetic diversity [18]. While *cox1* gene showed high resolution for resolving some algal species, the presence of introns in *cox1* in many species limits its application [16,35]. Molecular markers with high resolution and high specificity for intra-species genetic diversity analysis have been developed through comparative analysis of mtDNAs [19,21,22].

Our analysis revealed the mtDNA of *C. granii* was 34,970 bp in size, which was shorter than the mtDNA (36,071 bp) of *C. wailesii* and most diatom mtDNAs that were generally compact with few repeats and a small number of introns [36]. In this study, the mtDNA of *C. granii* included small intergenic regions, the absence of introns, and the low repeat content, which made it smaller size of mtDNA than others [22,37,38,39]. Rearrangements of DNA sequence containing four main events: translocation, inversion, duplication, and deletion [40]. Comparative analysis of *C. granii* and *C. wailesii* mtDNAs showed that most gene arrangements were conserved, except for the rearrangements of a few gene blocks, including translocation and inversion events. Notably, ribosomal protein coding genes *rps2*, *rps10*, and *rpl5* that are found in mtDNAs of many other diatom species [21,22] were absent from the mtDNA of *C. granii*. Furthermore, the two *orfs* identified in *C. granii* shared no similarity with these three genes. The reason for this may be that the three genes were lost or transferred to the nucleus during evolution [41]. The AT content of the mtDNA of *C. granii* was 75.70%, which was essentially the same as that of *C. wailesii* (75.00%), and substantially higher than that of *M. undulata* (78.40%) in the class Coscinodiscophyceae [40]. Comparative analysis of mtDNAs can facilitate exploring synteny relationships among closely related species and ascertaining gene gains or losses in evolution [42].

Mitochondrial genomes evolve quickly and thus provide a rich source of molecular variations at the nucleotide level, which made it more suitable for the development of high-resolution molecular markers [43,44]. In this study, we collected and isolated six *C. granii* strains from Jiaozhou Bay in China, covering two seasons (summer and autumn). We successfully developed a new molecular marker named *cgmt1* based on comparative analysis of *C. granii* mtDNAs, which showed high resolution and specificity in distinguishing different *C. granii* strains. Furthermore, we used *cgmt1* as the molecular marker to track and probe *C. granii* geographical distribution and genetic diversity in environmental samples collected in the Bohai Sea, the North Yellow Sea, the Jiaozhou Bay, the Changjiang Estuary, the East China Sea, the Pearl River Estuary, and the Beibu Gulf. In this study, phylogenetic analysis of *cgmt1* amplified from different samples showed that *C. granii* was mainly distributed in the Bohai Sea and the North Yellow Sea of China, which was consistent with the results of previous studies [12]. The results also highlighted the reliability and practicability of the newly developed molecular markers and indicated that *C. granii* had high genetic diversity. Indeed, such effective marker *cgmt1* could facilitate studies on *C. granii*. On the one hand, it could be used in metabarcoding analyses for tracking the geographical distribution patterns of *C. granii* genotypes not only in Chinese coastal waters but also other ocean regions of the world. On the other hand, we could distinguish different strains of *C. granii* from different geographical spans and seasons, especially during red tides period, to evaluate strain-specific differential contribution to red tides and to trace the possible origin of *C. granii* red tides.

## 5. Conclusions

Through comparative analysis of mtDNAs, we successfully developed a new molecular marker *cgmt1* (*C. granii* mitochondrial 1) with high resolution and specificity for *C. granii* intra-species genetic differences. We demonstrated that it could be used effectively to investigate the biogeographical distribution and track genetic diversity of *C. granii*. This molecular marker holds great potential applications for the studies on biogeographic distribution and intra-species genetic diversity of *C. granii* in coastal waters of China and other countries.

## Figures and Tables

**Figure 1 microorganisms-10-02028-f001:**
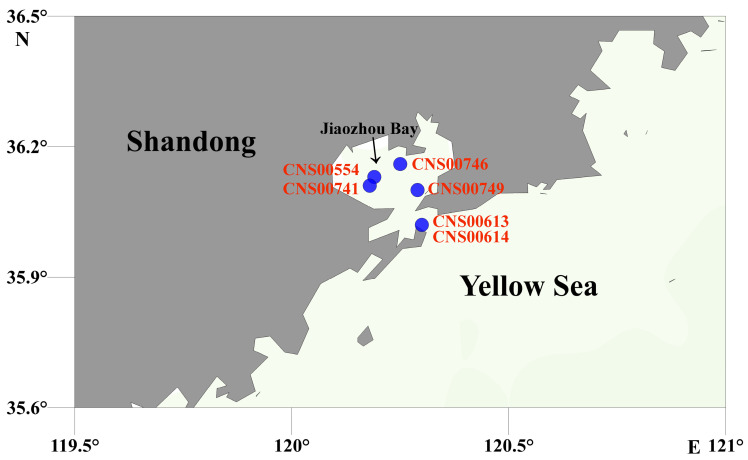
Sampling sites of the six strains of *Coscinodiscus granii.* Sampling sites are marked by blue shapes.

**Figure 2 microorganisms-10-02028-f002:**
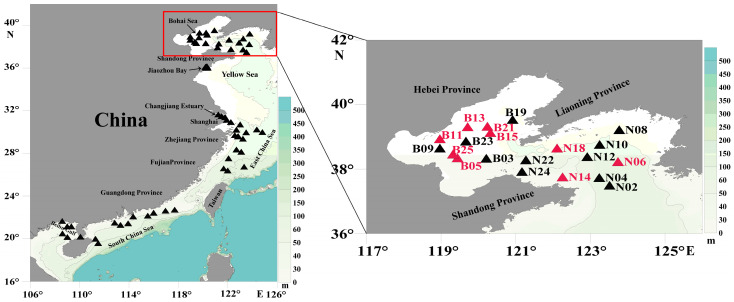
Geographical distribution of environmental samples used in this study. The environmental samples are marked by triangles. The results of positive amplified samples are marked by red shapes.

**Figure 3 microorganisms-10-02028-f003:**
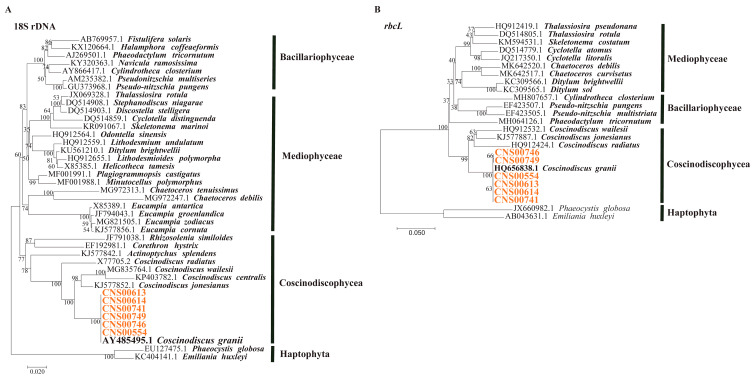
Identification of six strains of *C. granii*. (**A**,**B**): ML Phylogenetic trees based on the marker 18S rDNA and *rbcL*, respectively.

**Figure 4 microorganisms-10-02028-f004:**
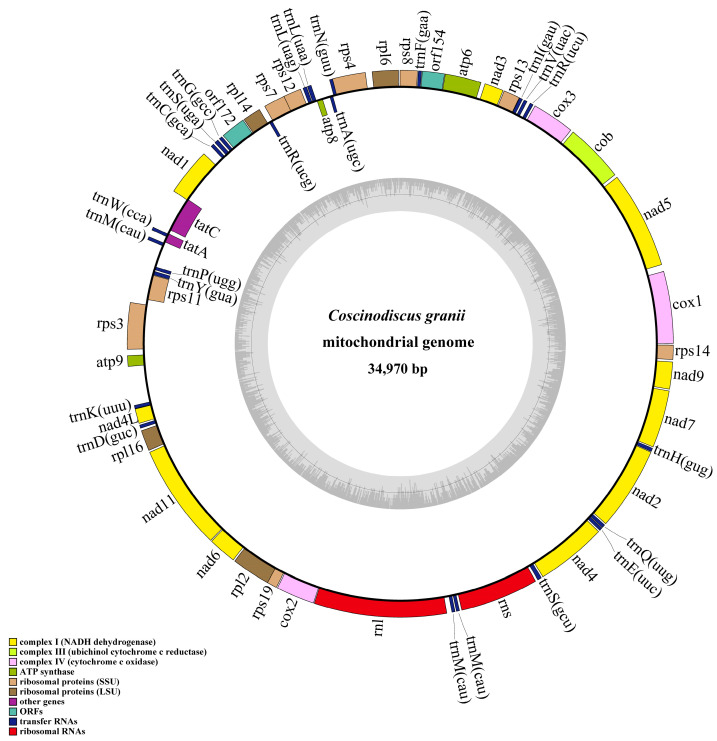
Circular maps of the mtDNA of *Coscinodiscus granii* (Strain CNS00554). Circular map of the mtDNA of *C. granii*. The transcriptional direction inside the ring is clockwise and the transcriptional direction outside the ring is counterclockwise. The protein-coding genes and rRNAs and tRNAs genes are labeled inside or outside the circle. The color of the gene boxes indicates the different functional groups of genes. The ring of bar graphs on the inner circle shows the GC content in dark gray.

**Figure 5 microorganisms-10-02028-f005:**
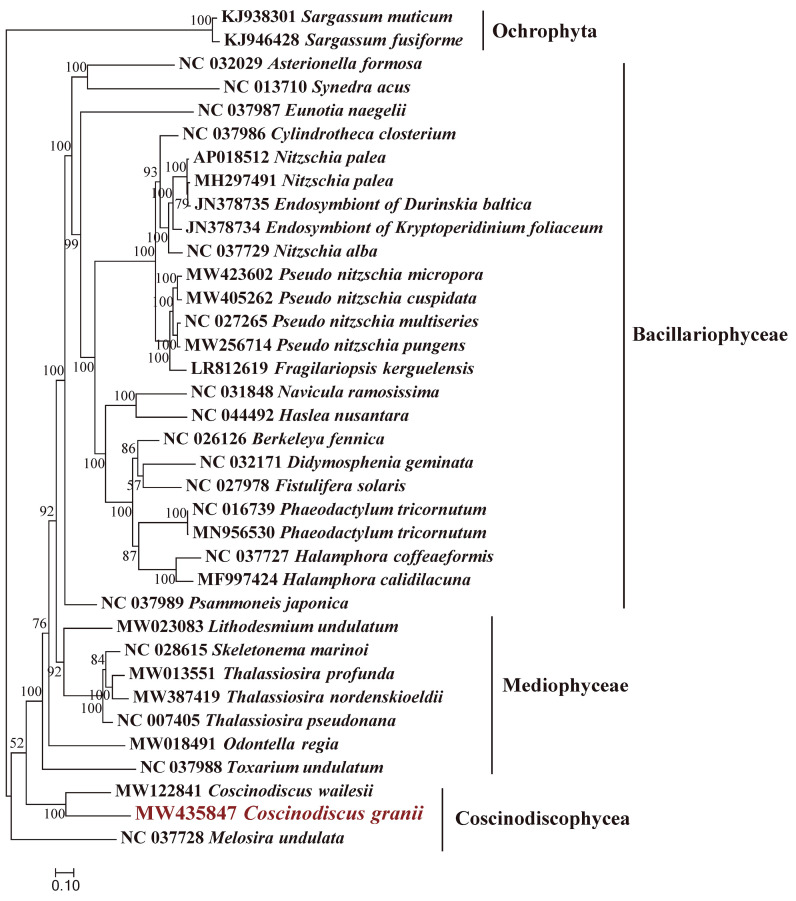
Phylogenetic analysis of *C. granii* (CNS00554). Maximum likelihood (ML) phylogenetic tree using concatenated amino acid sequences of 27 common protein-coding genes from 34 publicly diatom mtDNAs, and *Sargassum fusiforme* (KJ946428) and *Sargassum muticum* (KJ938301) in Ochrophyta were used as out-group taxa. The numbers beside branch nodes are the percentage of 1000 bootstrap values.

**Figure 6 microorganisms-10-02028-f006:**
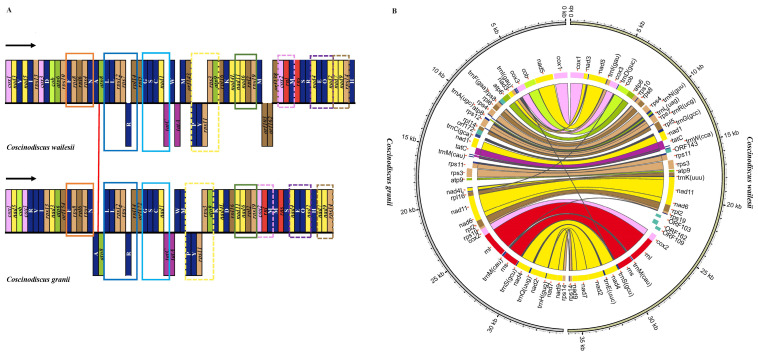
Synteny comparison of *C. granii* and *C. wailesii*. (**A**): Block diagram of gene content and gene rearrangement, tRNAs are indicated by capital single letter. Solid line boxes indicate the same conserved block of genes, while dotted boxes indicate translocation event and gene block connected by a red line indicates inversion event. Blocks with the same color represent the same type of genes. (**B**): The assignment of genes into different functional groups is indicated by colors.

**Figure 7 microorganisms-10-02028-f007:**
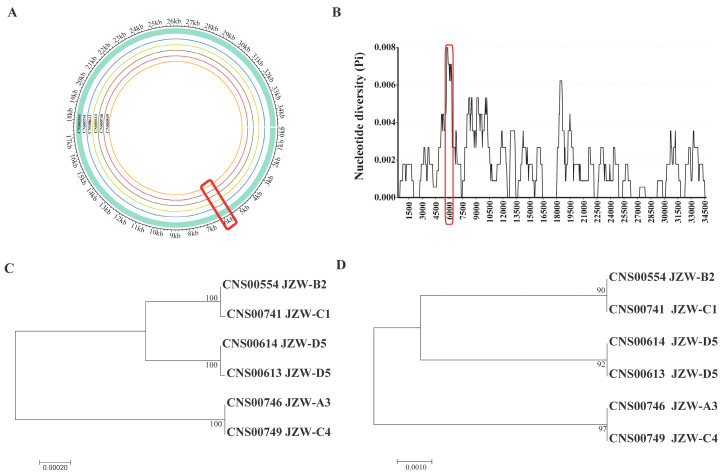
Genomic variations and phylogenetic trees based on maximum likelihood (ML) analysis of six *C. granii* strains. (**A**): Genomic variations density in six *C. granii* strains. The cyan band represented the reference genome CNS00554. The range of *cgmt1* region was marked by red box. From inside to outside, circles represent six *C. granii* strains isolated from the Jiaozhou Bay. (**B**): Sliding window analysis of nucleotide variability (Pi) across complete mtDNAs of six *C. granii* strains. (**C**): Phylogenetic analysis using the whole mtDNAs of six *C. granii* strains. (**D**): Phylogenetic analysis using the newly developed marker *cgmt1*. The sea area label marked with abbreviations: Jiaozhou Bay (JZW).

**Figure 8 microorganisms-10-02028-f008:**
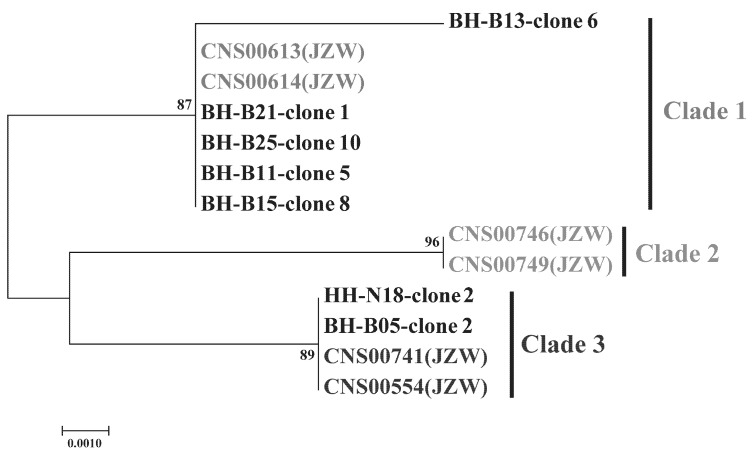
Phylogenetic analysis based on the *cgmt1* sequences of *C. granii* obtained from strains and environmental samples. The sea areas were marked with abbreviations: Jiaozhou Bay (JZW), Bohai Sea (BH), and Yellow Sea (HH).

**Table 1 microorganisms-10-02028-t001:** The sampling station information of the six *Coscinodiscus granii* strains.

Strain Number	Stations	Sampling Time	Longitude (°E)	Latitude (°N)
CNS00554	JZW-B2	2020.08	120.19	36.13
CNS00741	JZW-C1	2020.08	120.18	36.11
CNS00613	JZW-D5	2020.09	120.30	36.02
CNS00614	JZW-D5	2020.09	120.30	36.02
CNS00746	JZW-A3	2020.11	120.25	36.16
CNS00749	JZW-A4	2020.11	120.29	36.10

## Data Availability

The original contributions presented in the study are publicly available. The data can be found here: https://www.ncbi.nlm.nih.gov/nuccore/MW435847, MW122841, MZ571469-MZ57147, MZ561055-MZ561066, MZ544474-MZ544479, MZ544480-MZ544485 (accessed on 2 November 2021); Illumina raw sequences are available on NCBI’s sequence read archive (SRA) at BioProject PRJNA689860, PRJNA686182, PRJNA777367.

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
