# Peer review of "Genetic Diversity and Geographical Distribution of the Red Tide Species Coscinodiscus granii Revealed Using a High-Resolution Molecular Marker"

_microorganisms, 2022, doi:10.3390/microorganisms10102028_

Round 1

Reviewer 1 Report

This article reports a very narrow topic of ecological microbiology which is more detailed on Chinese marine environments (biogeography) than  on the molecular biology of the mitochondrial DNA studied for a single diatom species.

One would like to see a broadening of the narrow topic to similar toxic diatoms blooming around the world, at least in the Introduction and Conclusions. Moreover, the quality of some figures, in particular Fig. 3, is very poor. 

A substantial revision would be required for publication in Microorganisms and towards this end the Authors should consider the following specific points.

1.Why diatoms are not mentioned in the Abstract?

2. Why the available chloroplast DNA of the studied diatom  mentioned?

3. Intro line 53.-55 The text is very generic and written in poor English

4. The phylogenetic tree presented in Fig. 5 shows very compressed branches. Have the Authors considered to use a single protein marker with wide variation such as ND5 to strengthen their phylogenetic conclusions?

Author Response

Reviewer#1

This article reports a very narrow topic of ecological microbiology which is more detailed on Chinese marine environments (biogeography) than on the molecular biology of the mitochondrial DNA studied for a single diatom species.

One would like to see a broadening of the narrow topic to similar toxic diatoms blooming around the world, at least in the Introduction and Conclusions. Moreover, the quality of some figures, in particular Fig. 3, is very poor.

Response: Thanks for the reviewer’s suggestion. We have revised the Introduction and Conclusions. We have also improved the quality of the figures.

A substantial revision would be required for publication in Microorganisms and towards this end the Authors should consider the following specific points.

1.Why diatoms are not mentioned in the Abstract?

Response: We have added description of diatoms (at lines: 15-16): Diatoms are responsible for approximately 40% of the global primary photosynthetic production in marine ecosystems and account for up to 20% of global carbon fixation.

  1. Why the available chloroplast DNA of the studied diatom mentioned?

Response: We have removed from the manuscript description about chloroplast DNA.

  1. Intro line 53.-55 The text is very generic and written in poor English

Response: We have rewritten the sentence in the revised manuscript.

  1. The phylogenetic tree presented in Fig. 5 shows very compressed branches. Have the Authors considered to use a single protein marker with wide variation such as ND5 to strengthen their phylogenetic conclusions?

Response: We have modified Fig 5. Furthermore, in this study, the Maximum likelihood (ML) phylogenetic tree using concatenated amino acid sequences of 27 common protein-coding genes from 34 publicly diatom mtDNAs, and Sargassum fusiforme (KJ946428) and Sargassum muticum (KJ938301) in Ochrophyta were used as out-group taxa. The numbers beside branch nodes are the percentage of 1000 bootstrap values. We think our method with wide variation is more convincing than those obtained by using a single protein marker.

Reviewer 2 Report

Review of “Genetic diversity and geographical distribution of the red tide species Coscinodiscus granii revealed using a high-resolution molecular marker.” The manuscript is a fine contribution to the knowledge of red tide forming species C. granii, which aims to develop molecular markers with high resolution and specificity for C. granii by comparative analysis of mtDNAs, attempting to identify different taxa of this species. The manuscript is mostly well-written and contains relevant information. There are only some minor comments that need to be addressed as below when preparing the revised version. I am recommending to accept it with minor changes.

1. Please briefly describe the method and related software used for mitochondrial genome annotation in Materials and Methods.

2. Results-3.3-“---we carried out PCR assays using primers designed against cgmt1 and DNA preparations extracted from different 20 representative eukaryotic algae species---The results of all of these PCR reactions on these species were negative, further conforming high specificity of cgmt1 (Fig. S1)---”. Only 14 species are shown in the results (Fig. S1), please correct. In addition, in my opinion, No. 3 and No. 6 may not be negative, please explain.

3. Results-3.4-“Among these 66 PCR amplicons, nine showed strong signals”- Whether the authors have further validated the molecular results by microscopy observations.

4. Results-3.4-“Phylogenetic analysis based on cgmt1 sequence from six different strains of C. granii and 55 sequencing results from environmental samples showed high genetic diversity (Fig. 8).” 55 or 9 please check.

5. Did the authors assemble and annotate each of the six sample mitochondrial genomes? Or only the sample CNS00554, please clarify in the materials and methods in the text, and indicate which sample is selected in Figure 5 and Figure 6.

Author Response

Reviewer#2

Review of “Genetic diversity and geographical distribution of the red tide species Coscinodiscus granii revealed using a high-resolution molecular marker.” The manuscript is a fine contribution to the knowledge of red tide forming species C. granii, which aims to develop molecular markers with high resolution and specificity for C. granii by comparative analysis of mtDNAs, attempting to identify different taxa of this species. The manuscript is mostly well-written and contains relevant information. There are only some minor comments that need to be addressed as below when preparing the revised version. I am recommending to accept it with minor changes.

Response: Thanks for the reviewer’s suggestion. Those comments are all valuable for improving our manuscript. We have addressed all comments and have revised our manuscript accordingly. Revised details are marked in red in the revised version of our manuscript.

  1. Please briefly describe the method and related software used for mitochondrial genome annotation in Materials and Methods.

Response: We have provided more information in the revised manuscript as follows (at lines: 173-178): The circular mtDNA (CNS00554) of C. granii was 34,970 bp in size (GenBank accession number: MW435847). The annotation of protein coding genes (PCGs), transfer RNA (tRNA) genes, ribosomal RNA (rRNA) genes, and open reading frames (orfs) was conducted using Open Reading Frame Finder (ORF finder) with SmartBLAST and BLASTP, tRNAscan-SE and MFannot [9].

  1. Results-3.3-“---we carried out PCR assays using primers designed against cgmt1 and DNA preparations extracted from different 20 representative eukaryotic algae species---The results of all of these PCR reactions on these species were negative, further conforming high specificity of cgmt1 (Fig. S1)---”. Only 14 species are shown in the results (Fig. S1), please correct. In addition, in my opinion, No. 3 and No. 6 may not be negative, please explain.

Response: We have revised the description. We tested the specificity of the new molecular marker on 14 representative eukaryotic algae species. We used six C. granii strains as positive controls. While most of these 14 representative eukaryotic algae species showed completely negative results, some showed some but greatly reduced signals.

  1. Results-3.4-“Among these 66 PCR amplicons, nine showed strong signals”- Whether the authors have further validated the molecular results by microscopy observations.

Response: We have observed the water samples under microscope and identified phytoplankton species. However, different species of the genus Coscinodiscus could not be distinguished based on morphological features alone.

  1. Results-3.4-“Phylogenetic analysis based on cgmt1 sequence from six different strains of C. granii and 55 sequencing results from environmental samples showed high genetic diversity (Fig. 8).” 55 or 9 please check.

Response: We have rewritten the sentence in the revised manuscript as follows (at lines: 326-328): Phylogenetic analysis based on cgmt1 sequence from six different strains of C. granii and 55 sequencing results from nine positive environmental samples showed high genetic diversity (Fig. 8).

  1. Did the authors assemble and annotate each of the six sample mitochondrial genomes? Or only the sample CNS00554, please clarify in the materials and methods in the text, and indicate which sample is selected in Figure 5 and Figure 6.

Response: We have provided the information in the materials and methods in the text and Figures in the revised manuscript.